# Lifestyle change accelerates epigenetic ageing in King penguins

Robin Cristofari [1] ✉, Leyla R. Davis[2], Gaël Bardon[3,4], Flávia A. Nitta Fernandes[1], Maria Elena Figueroa[5], Sören Franzenburg [6], Michel Gauthier-Clerc[7], Francesco Grande[5], Richard Heidrich[5], Mikaela Hukkanen [8,9], Yvon Le Maho [4], Miina Ollikainen [8,9], Elodie Paciello[4], Patrick Rampal[3], Nils C. Stenseth [10], Emiliano Trucchi[11], Sandrine Zahn [4], Céline Le Bohec [3,4,12,14] & Britta S. Meyer [13,14]

A growing body of evidence supports the role of nutrient sensing and metabolism pathways in regulating ageing rate and healthspan, but the diversity of human lifestyles challenges our ability to identify the mechanisms of this age acceleration. Here, we examine how the transition of wild King penguins to zoo husbandry can closely mimic the shift to a Western lifestyle in humans, and shed light on conserved epigenetic changes in responses to sedentary conditions. We show that, just like modern humans, zoo-housed male King penguins experience an extended lifespan, but this comes at the cost of accelerated epigenetic ageing throughout life. This accelerated ageing is associated with differential methylation in key growth and maintenance pathways, including the mTOR and PI3K/Akt networks. Our results demonstrate the conserved link between lifestyle and age acceleration. Such evolutionary evidence may help us to improve risk detection and, ultimately, therapeutics for lifestyle-induced age acceleration in humans.

At the epidemiological level, there is a clear association between sedentary behaviour, obesity, and accelerated ageing phenotypes in humans[1,2]. Clinically, this association is reflected in the convergence of processes that include the dysregulation of nutrient sensing, metabolic disruption, impaired mitochondrial function, and a chronic inflammatory state associated with cellular senescence[1,3]. Within the nucleus, this translates into telomere shortening, associated with increased epigenetic drift[4,5], accelerated clock-like epigenetic changes[4,6], and decreased genome stability[1]. In short, sedentary behaviour and obesity accentuate most, if not all, of the hallmarks of ageing[7,8], and in humans, this is directly reflected in epigenetic age

acceleration (EAA)[4,6,9]. Thus, the resulting effect on ageing seems primarily pathological, stemming from a general derangement of homoeostasis[1]. Ultimately, this means that an individual's lifelong history of nutrient intake and energy management, although difficult to estimate in humans, is likely to be a key driver of EAA.

These findings have naturally led to the idea that improving nutrient intake and physical activity might be a shortcut to lifespan extension. Indeed, calorie restriction (CR)[5,10–12], physical activity[13], as well as direct manipulation of nutrient-sensing pathways[14,15], have shown positive effects on ageing rates. However, it is still unclear whether such benefits add up over a lifetime[16]: the spectacular benefits

[1]Institute of Biotechnology, HiLIFE, University of Helsinki, Helsinki, Finland. [2]Zoo Zürich, Zürich, Switzerland. [3]Centre Scientifique de Monaco, Monaco, Monaco. [4]Université de Strasbourg, CNRS, IPHC UMR 7178, Strasbourg, France. [5]Loro Parque, Tenerife, Spain. [6]Competence Centre for Genome Analysis, Kiel, Germany. [7]University of Geneva, Faculty of Sciences, Geneva, Switzerland. [8]Institute for Molecular Medicine Finland, HiLIFE, University of Helsinki, Helsinki, Finland. [9]Minerva Foundation Institute for Medical Research, Helsinki, Finland. [10]Centre for Ecological and Evolutionary Synthesis (CEES), Department of Biosciences, University of Oslo, Oslo, Norway. [11]Department of Life and Environmental Sciences, Marche Polytechnic University, Ancona, Italy. [12]CEFE, Université de Montpellier, CNRS, EPHE, IRD, Montpellier, France. [13]Research Unit for Evolutionary Immunogenomics, Department of Biology, University of Hamburg, Hamburg, Germany. [14]These authors contributed equally: Céline Le Bohec, Britta S. Meyer. ✉e-mail: robin.cristofari@helsinki.fi

of CR in the laboratory often disappear in a natural environment[17], while its "hidden costs" become more apparent[18]. Studies conducted in mice are limited by the species' short lifespan and characteristic metabolism[19], making them a limited model for examining some mechanisms in humans[20]. Clinical trials are scarce[21], and reliable long-term data dietary intake in humans extremely difficult to obtain[22], especially as modern food sources—in particular ultra-processed foods—are increasingly thought to act on ageing independently of their calorie content[23]. As a result, there is still fierce debate about whether the health span benefits observed in the laboratory also apply to natural, lifelong, periodic CR in a long-lived species such as humans[20,21,24].

Here, we test the effects of lifelong manipulation of physical activity and food intake on ageing rate using a novel model, the king penguin (*Aptenodytes patagonicus*). *Aptenodytes* penguins have long been studied for their outstanding fasting abilities, and are able to go without food for up to eight weeks—a fast that, despite its extreme length, shares key physiological traits with human fasting[25]. As part of their breeding cycle, King penguins alternate between prolonged CR and intense physical activity, with foraging trips requiring them to swim up to 1200 km in the open ocean in a few days[26]; and unlike any other model to date, penguins voluntarily stop eating in the wild, removing one of the current barriers standing between laboratory and real-life conditions[17,18].

For larger, long-lived wild animals such as king penguins, the transition to zoo husbandry has relevant similarities to modern Western human lifestyle. Protection from predators, reliable food sources and advanced veterinary care reduce extrinsic mortality hazards and can increase average lifespan[27]. On the other hand, despite best husbandry practices, physical activity is necessarily reduced. Indoor housing is often required, entailing artificial lighting and simulated seasonality. While these conditions are theoretically beneficial overall for longevity, depression-like emotional states can follow[28], and overweight is a recurrent issue in zoo-housed animals, often requiring specific dietary management[29]. For a species such as the king penguin, this creates a unique experimental system in which CR and high energy expenditure are effectively a control state, and sedentary behaviour and continuous feeding are an experimental manipulation.

Specifically, we hypothesise that introducing a Western-human-like evolutionary mismatch by inducing lifelong sedentary behaviour and increased food availability would lead to accelerated ageing in zoo-born king penguins. A candidate mechanism for this is the chronic activation of pro-growth pathways, and the associated suppression of cellular repair and autophagy mechanisms, e.g. via counter-acting properties of the mTOR pathway and other mechanisms balancing growth and somatic maintenance. If observed, such regulatory changes associated with EAA would independently validate the currently hypothesised impact of lifestyle changes on ageing rate, and would provide novel evidence for a robust, evolutionarily-conserved link between nutrient regulation, physical activity and ageing.

In this work, we show that zoo-housed King penguins display accelerated epigenetic ageing compared to their age-matched wild counterparts, from young ages until late life. Paradoxically, we show that this increase in EAA comes with an increase, and not a decrease, in life expectancy. We suggest that both increased EAA and increased life expectancy are two opposite-direction outcomes of the sheltered, well-fed, low-activity and predator-free lifestyle at the zoo: consistently with this, we show that age-independent epigenetic differences between zoo and wild birds concentrate in pathways related to adaptation to their lifestyle, including in particular metabolism and growth, physical activity, and circadian rhythms. This result suggests that a Western-like lifestyle change may have the double effect of increasing lifespan, but not healthspan - an effect in tune with current observations in humans.

## Results

### Epigenetic age is accelerated in zoo-housed penguins

To estimate biological ageing in king penguins, we relied on a CpG methylation-based epigenetic clock, as this approach has been effective at predicting frailty and mortality risk in a wide range of species[30,31]. In order to calculate EAA in king penguins in the absence of an externally-validated epigenetic clock, we produced EM-seq[32] whole-genome sequencing data from whole peripheral blood for 64 known-age samples for male King penguins, 34 of them from the wild (Possession Island, Crozet archipelago, in the Indian sector of the Southern Ocean), and 30 born and housed at the zoo (10 at Zoo Zürich, Switzerland, and 20 at Loro Parque, Tenerife, Spain). After read processing, we obtained whole-genome CpG methylation data at an average median depth of 29.5X across samples (range: 20X to 41X). After pooling methylation calls for both strands, our filtered dataset contained a total of ~10,000,000 high-quality CpG sites (see Supplementary Table 3). We did not find any evidence for genetic divergence between the wild and zoo groups (see Supplementary Methods 3), in keeping with the small number of generations elapsed since introduction from the wild (7 generations at most in our sample). We therefore exclude a systematic genetic basis for EAA in our experimental setup.

To infer epigenetic age, we used the residuals from an elastic net regression fitted to the principal components of variance in CpG methylation levels[31,33]. The final inter-group difference was highly significant when modelling predicted epigenetic age against calendar age: correcting for library conversion efficiency and genome-wide average CpG methylation level, the effect of living condition on average EAA at the zoo is estimated as 6.48 years (SE = 0.74, *p*-value < 0.0001). When allowing for separate slopes, the difference in slope was not significantly different from zero (−0.12, SE = 0.08, *p*-value = 0.16). This intercept-only difference is consistent with the currently accepted model of epigenetic ageing, under which CpG methylation at age-related sites is logarithmic, and not linear, with time[34] (see Supplementary Methods 4). Our final model, therefore, included only inter-group difference in intercept (baseline age acceleration), and had an overall excellent fit (RMSE = 2.48 years, $R^2$ = 0.91) (Fig. 1A, B). In that model, EAA was not different between individuals from the two different zoos included in the study, demonstrating that the difference observed in EAA is driven by lifestyle rather than the population sampled (*t*-statistic = −0.15, *p*-value = 0.88) (see Methods and SI-S4 for further validation of this result).

In order to control for possible overfitting effects, we fitted a similar PC-elastic net model on one group only (zoo or wild birds), before predicting EA values for the other group according to this model. The inter-group differences were calculated using a linear model with the lifestyle group as a fixed factor: it is inferred to be 6.32 years (SE = 1.05, *p*-value = 1.02e⁻⁷) when training on zoo birds, and 3.83 years (SE = 0.71, *p*-value = 1.36e⁻⁶) when trained on wild birds (see Supplementary Fig. 7). Alternatively, we also constructed a more direct estimator of epigenetic age as the unweighted average of CpG methylation across the 9839 CpGs showing the strongest negative Pearson's correlation with age in our sample (with a threshold of $R^2 \geq 0.2$), without any additional covariates (age-related differentially-methylated sites, aDMS). The average methylation value across aDMS sites was linearly scaled to the observed range of ages in our data, and linearly modelled as a function of log(age) (Fig. 1C). While this estimator is noisier than the elastic-net based approach, it is entirely independent from sample origin at every step of the estimation. A linear model regressing this aDMS age against calendar age and origin (zoo or wild), correcting for library conversion efficiency and genome-wide average CpG methylation level, also supports significantly accelerated ageing at the zoo, by 2.63 years (SE = 0.51, *p*-value < 0.00001), see Fig. 1D. The numerical differences between these approaches may be explained by different

sensitivities (training sample size is halved in the first of these controls, and the second one is based a cruder averaging procedure that gives equal weights to all included aDMS CpGs, regardless of the strength of their actual covariance with age). The true value of the age acceleration at the zoo is thus likely between 2.63 years (the most conservative estimate) and 6.48 (the estimate derived from a PCA of the whole sample set)−in any case, we reach the conclusion that zoo-housed birds are significantly age-accelerated compared to their wild counterparts.

As an additional control, and in order to rule out Type II errors, we applied our approach to a human test dataset of similar characteristics (see 'Methods' below), for which EAA values are known[35]. We used as a contrast the strongest single known lifestyle factor affecting EAA in humans: smoking[36]. While this stressor is not directly comparable to a general lifestyle change, it allows us to test for model behaviour in a well-known system with clear expectations. Our approach for EAA in smokers versus non-smokers yielded results that were highly consistent with the independently-trained PCPhenoAge and PCGrimAge clocks[33] (see Supplementary Methods 4 and Supplementary Fig. 8). Importantly, as in the King penguin case, regression lines differ only in intercept, and not in slope, as expected from the logarithmic model of ageing[34]. These results strongly support the validity of our modelling approach, and the conclusion that king penguins housed in zoos display higher levels of EAA than their wild counterparts: relative to the

species longevity (~40 years), this difference in EAA is comparable in magnitude to the extremes observed in humans, here between smokers and non-smokers, and make the zoo transition a major driver of EAA.

## Zoo-housed penguins have increased survival

In contrast with the observed age acceleration in zoo conditions, survival analysis using Cox proportional-hazard modelling showed that zoo-housed penguins lived longer on average than their wild counterparts, with nearly no differences between sexes (Fig. 1E, F). This analysis included 1895 wild and 305 zoo-housed king penguins. For zoo-housed birds, the median survival age (with 95% confidence intervals) was 20.7 years [18.7–23.6] for males and 20.8 [18.8–23.7] for females. In comparison, the median survival age for wild birds was significantly lower, at 13.5 years [13.0–14.1] for both sexes. The hazard ratio comparing zoo-housed to wild conditions was 0.46 [0.38−0.55], indicating a substantial advantage for penguins housed in zoos (the methodology and additional analyses can be found in the Methods section and SI-S2). Age-specific survival is lower at younger ages in the wild, reflecting the strong effect of extrinsic mortality (predation and starvation at sea in particular), while protection from accidental death, food security, and a high level of medical care shelters zoo-housed penguins until more advanced ages - an observation consistent with previous results in zoo-housed mammals[27]. As a result, at equal

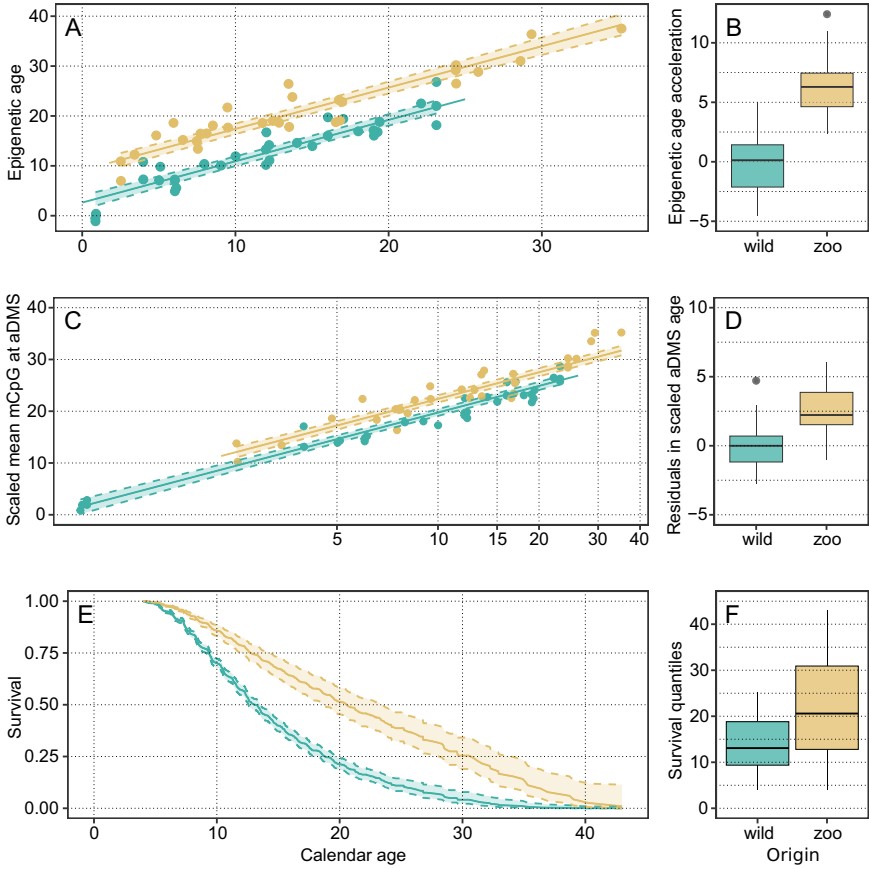

**Fig. 1 | Epigenetic age acceleration and survival in wild and zoo-housed King penguins. In green, wild birds, in tan, zoo birds; boxplots provide color reference. A** Comparison of epigenetic age versus calendar age between wild and zoo-housed King penguins, linear regression with best-fit line and 95% confidence intervals. Data points represent raw age predictions derived from CpG methylation levels. **B** Distribution of age acceleration in wild (N = 34) and zoo-housed (N = 30) King penguins (center line, median; box limits, upper and lower quartiles; whiskers, 1.5x interquartile range; points, outliers). **C** Scaled average methylation level at aDMS sites, linear regression with best-fit line and 95% confidence intervals,

regressed against log(calendar age). **D** Distribution of residuals in wild (N = 34) and zoo-housed (N = 30) King penguins (center line, median; box limits, upper and lower quartiles; whiskers, 1.5x interquartile range; points, outliers). **E** Cox proportional hazard (Cox-PH) model illustrating survival probabilities for wild and zoo-housed male King penguins. Error bands figure the 95% confidence interval bounds for the hazard ratio for each stratum. **F** Survival quantiles (median, quartiles and minimum-maximum values) from the Cox-PH model for 34 wild and 30 zoo-housed King penguins. Source data are provided as a Source data file.

calendar age, survival is considerably increased at the zoo compared to the wild.

We examined whether selective disappearance was a likely cause of this difference, with the proportion of frail and age-accelerated individuals decreasing more rapidly through time in the wild. However, the absence of correlation between EAA and age, either in the wild (Spearman's rank test $p$-value = 0.28), in the zoo (0.13), or in both (0.52), strongly suggests that selective disappearance is not a major determinant of EAA in either condition, as strong survival-bias would induce a change in age-class individual composition. Yet, all the individuals included in this study were still alive, so that direct association between age acceleration and lifespan is, at this stage, impossible. Overall, we conclude that birds housed in zoo conditions undergo accelerated ageing, but that this age acceleration is compensated by their highly protected lifestyle, allowing for survival to advanced ages in conditions of frailty, which would rule out survival in the wild.

### Differentially methylated genes

In order to understand the determinants of age acceleration at the zoo, we searched for differentially methylated regions (DMRs) between zoo-housed and wild birds independently of age-acceleration signal. We beta-corrected individual CpG methylation levels[37] for chronological age, EAA and genome-wide methylation level (see Methods), in order to only retain CpG methylation signal that is consistently different between groups, but does not covary with either chronological age or with EAA. This approach corrects methylation rate at the CpG level: while it does not correct for possible complex, non-linear aspects of the ageing process, which may vary qualitatively between wild and zoo conditions, it does remove the linear components of variance associated with calendar age and with EAA at each locus. Thus, DMRs inferred from this signal should reflect specific methylation changes that oppose zoo and wild penguins in a time- and EAA-independent manner. This approach identified a total of 600 candidate DMRs (Fig. 2A), clustered in or near 292 known genes, and

36 predicted open reading frames (ORFs) in the king penguin—a result well outside random expectations (see Supplementary Methods 5 and Supplementary Fig. 11).

### Overrepresentation of growth and maintenance pathways

Pathway overrepresentation, tested against the Reactome pathway database[38], clustered these genes into 11 FDR-corrected 'super-paths' (See Supplementary Table 4 for the full results). Our gene set was significantly enriched for pathways centrally involved in cell growth, and in the coupling of nutrient sensing to ageing and age acceleration[39–41]. These include (1) the MAPK and RAS-RAF-MEK-ERK pathways (18 genes, FDR-corrected $p$-value = 0.0003), (2) the PI3K/Akt network (7 genes, 0.0013), (3) Notch signalling (4 genes, 0.0012), (4) ALK signalling (5 genes, 0.0049), (5) EGFR signalling (5 genes, 0.0008), and (6) the Rho GTPase Ras subfamily (4 genes, 0.0042). These pathways are consistently involved in (a) the regulation of cell growth (including mTORC2 subunit *MAPKAP1*, mTOR regulator *SIK3*, the platelet-derived growth factor receptor A *PDGFRA*, or the fibroblast growth factor receptor 3 *FGFR3*[42,43]), (b) apoptosis (including p53 effector *PERP*, proto-oncogene *REL*, apoptosis-inducing factor mitochondria-associated *AIFM3*, and both *EVA1B* and *EVA1C*, the two known paralogs of the regulator of programmed cell death *EVA1A*), and (c) DNA-damage response (e.g. *HSF1*, *ZGRF1*, *BUB1*, *SSBP2*, *SLX4*, *GADD45A*, *WRD76* or *USP28*)—see Fig. 2B for a representative subset of these genes, and the Source Data file for the full list of results. The clear overrepresentation of central growth and maintenance pathways in the EAA-independent differential methylation between zoo and wild conditions strongly supports our hypothesis and suggests that the induction of a sedentary, well-fed lifestyle has a direct influence on core metabolic regulation in king penguins.

### Change in diet, heart function and physical activity

Our results include DMRs associated with genes directly involved in coping with excessive nutrient intake, including *INPP4B*[44] and *ASIC2*[45]

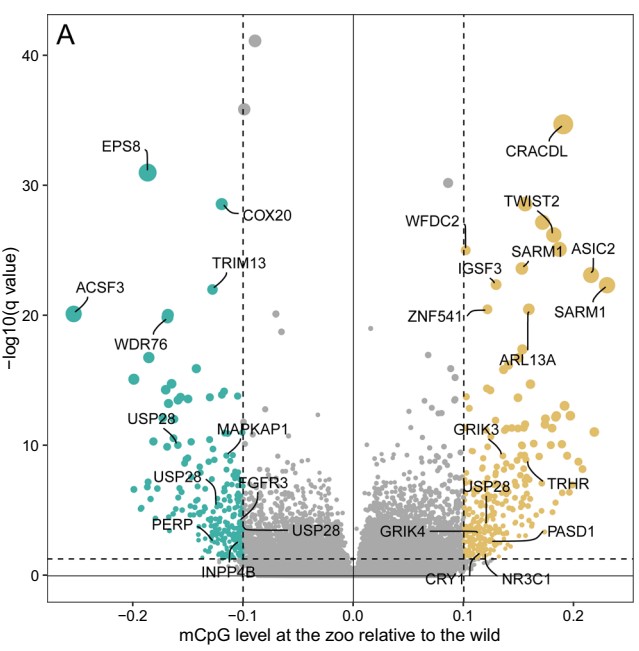
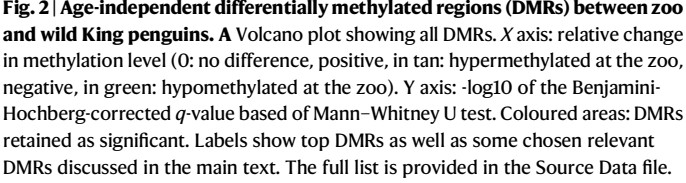
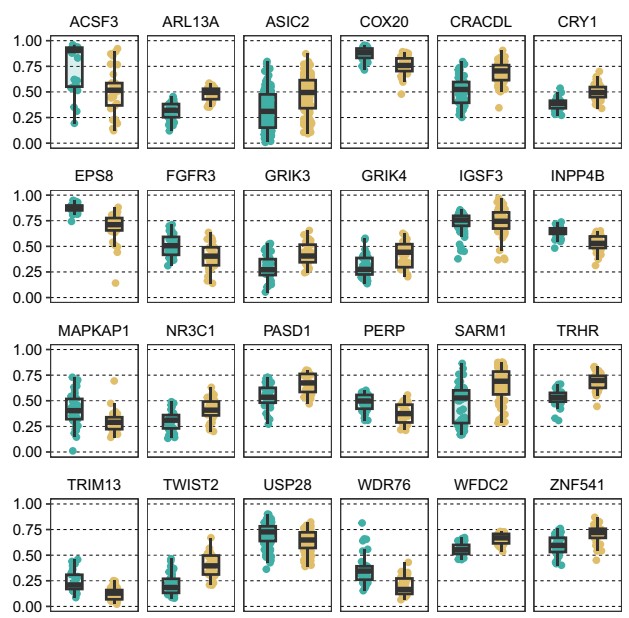

**Fig. 2 | Age-independent differentially methylated regions (DMRs) between zoo and wild King penguins. A** Volcano plot showing all DMRs. *X* axis: relative change in methylation level (0: no difference, positive, in tan: hypermethylated at the zoo, negative, in green: hypomethylated at the zoo). Y axis: -log10 of the Benjamini-Hochberg-corrected *q*-value based of Mann–Whitney U test. Coloured areas: DMRs retained as significant. Labels show top DMRs as well as some chosen relevant DMRs discussed in the main text. The full list is provided in the Source Data file.

**B** Adjusted methylation level at the DMRs labelled in the volcano plot. In green (left), wild birds ($N$ = 34), in tan, zoo birds ($N$ = 30); boxplots provide color reference. Methylation levels are beta-corrected for age, epigenetic age acceleration and genome-wide mean CpG methylation. Boxplots: center line, median; box limits, upper and lower quartiles; whiskers, 1.5x interquartile range. Individual points are shown. Source data are provided as a Source data file.

and NAD hydrolase *SARM1* - the latter centrally involved in cellular response to glucose, growth and apoptosis, whose deletion specifically improves function in the diabetic heart[46] and mitigates the effects of high-fat diets[47]. Interestingly, several of the DMRs in our study also overlap with genes associated with heart function and physical exercise in a recent genome-wide association study (GWAS) in a large human cohort[48], including *GRIK3* and *GRIK4* (the first subunit of this receptor, *GRIK1*, is linked to heart rate increase and recovery in humans[49]). These overlaps also include *RNF220*, *KCNH5* (its paralog *KCNH8* is a GWAS in humans), and *CAVIN-1* (a main interactor of *CAV1* identified as a GWAS). These convergences occur at different levels (we are focusing here on differentially methylated genes, and the Rhineland cohort study cited[48] focused on DNA polymorphism), but both types of evidence support the involvement of similar genes in adapting to physical activity in humans and penguins. It is also remarkable that epigenetic changes in these genes can be reliably detected in blood, as opposed to tissues more centrally involved in diet and physical activity (e.g. muscle, liver or adipose tissue). This underlines the value of a non-invasive matrix such as peripheral blood for insights on whole-organism methylome[50]. These results primarily suggest that (1) zoo-housed King penguins may mobilise a wide array of genes to specifically compensate for a change in diet lipid composition (from high-fat myctophid fish[51] to, typically, *Culpea* herring) and/or mitigate the negative effects of food abundance and (2) the suppression of intense physical activity in a species adapted to sustained efforts, that can routinely swim several hundred to thousands kilometres in a few days[26], contributes directly to systemic epigenetic change. Further research on these genes may provide candidates for mitigating the effects of adiposity and sedentary behaviour in humans.

## Effects of life indoors
At the zoo, birds have a significantly hypermethylated DMR in the promoter of *NR3C1*, the glucocorticoid receptor (GCR), and another ~3 kb upstream of *TRHR*, the thyrotropin-releasing hormone receptor. In humans, increased *NR3C1* promoter methylation is associated with decreased GCR expression and psychosocial stress[52]; and it correlates negatively with circulating corticosterone levels in seagulls[53]. This suggests low situational stress and low nutritional stress (CG levels increase with CR in King penguins[25]). Hypermethylation upstream of *TRHR*, on the other hand, may be associated with lowered *TRHR* expression and a blunted thyroid-stimulating hormone (TSH) response to TRH, a trait regularly observed in depression in humans. The observed methylation patterns are thus consistent with a sheltered lifestyle with lower GC-mediated stress. Characteristically, the circadian clock pathway is also significantly enriched in our DMR results (5 genes, FDR-corrected p-value = 0.0039, including cryptochrome circadian regulator *CRY1* and *PASD1*, a known clock repressor[54]). These results strongly suggest that sedentary, indoor lifestyle has a broad impact on zoo-housed king penguin physiology that goes beyond the pure effect of sedentary behaviour, and also reflects changes in, e.g. circadian cycles, and feeding and sleeping patterns. While this disruption of life rhythm is independent from sedentary lifestyle, it is regularly associated with it, and is indeed another characteristic of the Western lifestyle[55]. A possible association with a depression-like state, on the other hand, remains speculative and should be investigated further—but acute stress is unlikely to be causal in this case.

## Discussion
Our results bring novel, independent evidence to support the idea that the transition to a sedentary, sheltered and food-stable lifestyle is a conserved cause of age acceleration across a wide evolutionary divide. Importantly, they suggest that EAA is driven by the suppression of periodic CR and of physical activity, and the general disruption of life rhythms, rather by than lipotoxicity or other direct pathological effects of overweight: indeed, penguins in this study were not clinically obese (at the zoo, inter-breeding body mass was 11.8 kg, sd = 2.2, not markedly different from a ~12 kg inter-breeding wild bird[25]). Instead, differential methylation analysis points to pathways deeply involved in the systemic regulation of metabolism, growth and maintenance, rather than at specific effects on e.g. obesity-induced inflammation.

Naturally, the changes associated with the transition to zoo conditions are broad and multifaceted, extending beyond reduced caloric intake and physical activity. While this study demonstrates the impact of this transition on ageing rate, and suggests that metabolic regulation is deeply involved, the precise mechanisms through which this impact happens remain to be explored - whether through CR, sedentary behaviour, disruption of life rhythms, reduced environment complexity, altered microbial environment, or other factors. Nevertheless, these results considerably strengthen the hypothesis that evolved metabolism mismatch, rather than side-effects of the associated cultural changes, is at the root of the ongoing increase in age-acceleration diseases in humans.

This study has three additional methodological limitations: first, we focus mainly on adult, breeding-age birds. Thus, we are not able to examine whether the apparent difference between wild and zoo-housed birds is established already in early life, or appears gradually. While the intercept difference between the two groups (Fig. 1A) is very consistent with logarithmic EA in the presence of a stable age-accelerating stressor (as developed in Supplementary Methods 4), we cannot exclude strong effects of early-life rearing conditions. Second, the sampling dates are not entirely standardised in our study relative to the onset of the breeding season. All samples have been collected during the breeding season, and dates do not significantly differ between groups: but the epigenome is known to change with the circannual cycle[56], and this incomplete standardisation may introduce some noise—although likely not a systematic bias—into our results. Third, we include two zoos, but only one wild population—as there is currently only one known-age wild King penguin population covering the whole age span of King penguins in the world[57]. This study thus lacks a fully independent biological replicate in the wild.

Finally, we underline the fact that the observed EAA does not appear to be the result of an overall pace-of-life acceleration, as breeding output at the zoo is also decreased compared to the wild[58]. This decoupling of age acceleration and pace-of-life suggests the existence of a large, unexploited margin for lifestyle-based lifespan extension: in the penguin model, a "best of both worlds" lifestyle combining the physical activity, natural feeding rate and slower ageing rate of wild birds, together with the reduction of extrinsic mortality and medical care of the zoo would, in theory, result in a 5–20% increase in survival, depending on the value retained for age acceleration. In humans, this would be equivalent to a ~4 to up to ~15 years gain in life expectancy—at best, more than the estimated years of life lost to severe obesity in the Western lifestyle[59]. While such an extrapolation from penguins to humans is speculative at best, it underlines the magnitude of the gains in life expectancy that could possibly be achieved by acting solely on lifestyle factors—in modern humans as well as in animals under their care.

## Methods
### Sample collection
Sampling complied with the relevant ethical regulations. DNA samples ($N = 64$; $N = 34$ from the wild and $N = 30$ from zoos) were extracted from the whole peripheral blood of male King penguins of known age (see Supplementary Methods 0 for details). The free-ranging individuals were followed since fledging in the wild (Possession Island, Crozet Archipelago), as part of a long-term monitoring project[57], whereas the individuals from zoos were hatched in European zoos, and housed at Zoo Zürich, Switzerland ($N = 10$), and Loro Parque, Spain ($N = 20$), at the time of sampling. In the wild, the sampling was

approved by the French ethics committee in Strasbourg (CREMEAS, APAFIS#29338-2020070210516365) and the French Polar Environment Committee (Comité de l'Environnement Polaire - CEP), and permits to handle the animals and access the breeding sites were issued by the 'Terres Australes et Antarctiques Françaises'. In zoos, samples were collected for routine veterinary examination as part of Zoo Zürich and Loro Parque's husbandry licences, their reuse was not subject to additional authorisation. The choice to restrict the analysis to males only was guided by the asymmetry in fasting behaviours between males and females in this species: males systematically take the first turn in incubating the egg, fasting for at least 6 weeks on average at the start of reproduction, while females go foraging at sea immediately after egg laying, staying on land for ~3 weeks. Males thus offered a higher contrast in fasting behaviour between wild and zoo conditions.

## Sequencing and data processing

Whole-genome enzymatic-conversion methylome sequencing libraries were prepared using the NEB EM-seq kit[32], and sequenced on 6 sequencing lanes on the Illumina NovaSeq 6000 platform at CCGA in Kiel, Germany. Raw reads were trimmed, removing leading and trailing bases if the quality score dropped below 3 and retaining only reads of length > 36 bp. We used BCREval[60] (GitHub commit version #4e74fa8) to assess conversion efficiency during library preparation based on false-positive non-CpG methylation calls in telomeric repeats. Reads were aligned to the King penguin's reference genome (GCA_010087175.1) using bsbolt[61] (v1.5.0), and deduplicated and filtered using samtools[62] (v1.16.1). Individual methylated cytosines were called using bsbolt. In vertebrates, DNMT1 activity ensures that CpG methylation is normally symmetric across strands[63]: therefore, we calculated per-CpG average cytosine methylation level by pooling C and T calls from both strands at each CpG dinucleotide. We masked from our dataset all known C/T (or G/A) SNP positions based on a curated database of 40 King penguin whole genomes. Mean conversion efficiency, as assessed through telomeric methylcytosine false-positive call rate, was high at 99.5% (range: 96.6% to 99.9%): this method yields results that are very close to the internal PhiX control, but that reflect more directly the conversion efficiency on the target species' DNA[60].

## Data annotation

The genome reference was re-annotated for this study. Although this reference is highly BUSCO-complete (C:95.6% [S:95.0%, D:0.6%], F:1.5%, M:2.9%, n:8338), the published annotation included a large proportion of incomplete gene models. We used BRAKER[64] v. 3 to generate genome annotations based on (i) whole-transcriptome data for the wild King penguin[65], aligned to the King penguin genome using HiSAT2[66] v2.2.1, and (ii) the OrthoDB[67] v12 vertebrate protein reference dataset. Untranslated regions (5'- and 3'-UTRs) were annotated using GUSHR (https://github.com/Gaius-Augustus/GUSHR, GitHub commit version #ee26d5c), based on the same sources of information. Cis-regulatory promoter elements were defined as the 1500 bp upstream and 500 bp downstream of the transcription start site (TSS). CpG islands (CpGi) were defined as regions of high CpG density using CpGCluster[68] (GitHub commit version #248b42d). Gene annotations were established using protein-protein BLAST[69] against avian proteins.

## Epigenetic age acceleration (EAA)

In order to assess epigenetic age acceleration in wild and zoo individuals, we followed the approach outlined by Higgins-Chen and colleagues[33], performing first a principal component analysis using MethylKit[70] v1.30.0 (Supplementary Fig. 4). However, we reduced the extremely large starting CpG set in two steps: (i) we retained only sites with an average single-stranded depth of 10 to 25x, and an individual sample depth of 5 to 50x, no missing data in any individual, and a standard deviation in CpG methylation level of at least 0.1 (2,670,062 CpGs), and (ii) to retain only broadly informative sites, we calculated Pearson's correlation coefficient between individual age and CpG methylation level at each CpG, and only retained sites with an $R^2 \geq 0.2$ and a $p$-value $\leq 0.05$ (10,205 CpGs). We used the resulting components of methylation variation as independent variables in an elastic net regression using glmnet[71].

Our hypothesis is that age acceleration is lifestyle-dependent: we therefore expect it to be bimodal for a dataset with individuals in equal parts from the zoo and the wild, which breaks model assumptions for elastic-net regression residual distribution. To overcome this limitation, we used a grid approximation approach to identify a separate best-fit slope and intercept for both groups. Namely, we divided the parameter space into an evenly spaced discrete grid, starting with possible slopes ranging from −1 to 1 in log2 space (half to twice the ageing rate for zoo compared to wild individuals) and possible intercepts (relative age acceleration) ranging from −15 to 15 years. We also tested elastic net α mixing parameter values ranging from 0 to 1, with parameter range and step size decremented iteratively.

For each grid value, we performed a leave-one-out fit: the age of zoo individuals was first transformed linearly using the local slope and intercept, and, taking each sample out in turn, we trained an elastic net model using all principal components of methylation variance using N-fold (LOOCV) internal cross-validation on the 63 remaining samples. In other words, for each point, the other 63 points are the training set, and LOOCV is performed among these 63 training points, before predicting EA for the 64th point. We then used this model to predict epigenetic age for the 64th sample. For each vector (*slope, intercept, α*), we computed the model fit as the $R^2$ coefficient of a least square model where predicted age was regressed against the transformed age for each sample. We repeated this procedure iteratively at finer grid steps around the grid maximum. Finally, to evaluate the relevance of the best-fit slope and intercept, we linearly modelled predicted age as a function of true calendar age and living conditions, including individual as a random effect to account for longitudinal sampling of a subset of birds, and evaluated the magnitude and significance of the living conditions effect using estimated marginal means. For each individual, age acceleration was calculated as the difference between predicted age and true age at sampling.

This approach to define EAA was validated in two independent ways. First, we applied it to a human dataset of similar characteristics and known age acceleration parameters, including Infinium 450 K BeadChip data for 64 human males of a wide range of ages, 32 of which were current smokers amongst the most age-accelerated quantile, and 32 never-smokers amongst the least age-accelerated quantile, as calculated using the independently-trained PCPhenoAge and PCGrimAge clocks[33]. We chose to focus on age-accelerated current smokers and age-decelerated non-smokers to maximise the chance that age acceleration difference between the two groups shared a physiological mechanism, in order to parallel our King penguin experimental design. The aim of this first test was to verify that our approach successfully differentiated between the two groups, and proposed age acceleration values that are consistent with well-established clocks (see Supplementary Fig. 8). Second, we repeated our approach 500 times on the King penguin dataset, randomising origin (wild/zoo) amongst individuals at each iteration, and evaluated the probability of the observed parameters against a normal distribution fit to the empirical random distribution. The aim of this second test was to test for false-positive inter-group differences (see Supplementary Fig. 5).

## Differentially methylated sites and regions

In order to identify differentially methylated sites (DMS) between wild and zoo-housed individuals, we fit per-site binomial models on observed methylated and unmethylated cytosine counts at filtered CpG sites, selected with an average depth of 10 to 25x, and an

individual sample depth of 5 to 50x, with no more than 20% missing data (10,147,333 CpG sites out of 18,850,653 in the King penguin's genome). Models were fitted using the lme4 package[72] v1.1.36 in R. For each model, we included as covariates the bird's living conditions (wild or zoo), age at sampling, epigenetic age acceleration, enzymatic conversion efficiency, and genome-wide mean methylation level. Identity of the bird was included as a random factor to account for repeated sampling in some individuals. Log2(odds ratio) for living conditions was extracted from each model. We used the well-proven Metilene algorithm[73] v0.2-8 to identify differentially methylated regions (DMR) —a choice guided by this conservative method's very low FDR when benchmarked against other algorithms[74]. To account for known confounding effects, however, we corrected methylation levels[37] prior to segmentation through beta regression using glmmTMB[75] v1.1.10, and the same covariates as above. We retained only sites that had a minimum standard deviation of 0.1 before beta-correction, and conservatively filtered out DMRs with an absolute difference ≤ 0.1 and/or including <10 CpG sites. Maximum distance between CpGs within a DMR was set to 500 bp, corresponding to twice the median inter-CpG distance in this dataset. *P*-values were Benjamini-Hochberg corrected with $\alpha = 0.05$. DMRs were annotated, retaining only genes (including cisregulatory element) at a maximum distance of 5 kilobases from the DMR boundary. In order to evaluate the risk of false-discovery despite BH FDR correction, we conducted thorough randomisation tests (see Supplementary Fig. 11). Pathway overrepresentation was assessed using FDR-corrected binomial tests against the Reactome pathway database, using the GeneAnalytics toolkit (geneanalytics.genecards.org)[76].

## Survival analysis

We obtained survival data for both wild and zoo King penguin populations, and calculated survival curves in both environments as described in full detail in Supplementary methods 2. Briefly, in the wild, we used capture-mark-recapture data for 1895 known-age individuals monitored electronically since 1998[57]. Based on this data, we determined that individuals which had not been detected for at least 2 consecutive years had a probability of later return of ~0.3%, and could therefore be considered dead: this information allowed us to build an empirical Bayes predictor for the posterior probability of survival. At the zoo, we collected data from the Species360 database (Species360 (2024), https://species360.org/). Data included all reliably registered King penguins held in zoos across the world since 1913 ($N = 305$). We retained only individuals with known hatching date, and either (i) known natural death date ($N = 174$) or (ii) known right-censoring (still alive) date ($N = 131$). Finally, we used a Cox proportional-hazard model to compare survival probability in the wild, and at the zoo (see Figs. 1C, D).

## Reporting summary

Further information on research design is available in the Nature Portfolio Reporting Summary linked to this article.

## Data availability

The raw DNA methylation data generated in this study have been deposited in the Sequence Read Archive under BioProject PRJNA1187342. Both raw and processed DNA methylation data, as well as complete sample metadata, are available from the Etsin repository (etsin.fairdata.fi) with https://doi.org/10.23729/fd-e1da6857-0370-3d49-b1c3-4b7c53b366cc. Source data are provided with this paper.

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

## Acknowledgements

This study was funded and supported by the Research Council of Finland (grants #331320 and #354649 to RC), by the French Polar Institute Paul-Emile Victor (IPEV Project 137-ANTAVIA, PI CLB), by the Centre Scientifique de Monaco (CSM) with additional support from the RTPI-NUTRESS (CSM/CNRS-UNISTRA), by the Centre National de la Recherche Scientifique (CNRS) through the Zone Atelier Antarctique et Terres Australes (ZATA), by Zoo Zürich, and by Loro Parque. This work was supported by the DFG Research Infrastructure NGS_CC (project 407495230) as part of the Next Generation Sequencing Competence Network (project 423957469). Sequencing was carried out at the Competence Centre for Genomic Analysis (Kiel). We especially acknowledge the key contribution of the Finnish IT Center for Science (CSC) for access to computational resources necessary to the realisation of this work. We also thank Prof. Jaakko Kaprio for providing data for humans used in this study, and Prof. Denis Allemand for his role in the conception of this project and his continued support. We are deeply grateful to Zoo Zurich and to Loro Parque teams for their expert support in planning and executing this experiment, as well as to all the wintering and summering members of IPEV Project 137 and all the other colleagues and students within the P137 team, who participated in the long-term monitoring and sample collection since 2000. We also sincerely thank the IPEV logistics teams for their important and continued support in the field. This study is part of and supported by the long-term Studies in Ecology and Evolution (SEE-Life) programme of the CNRS.

## Author contributions

The study was designed by R.C., B.M., and C.L.B. Sample collection was done by R.C., C.L.B., L.R.D., M.E.F., F.G., R.H., E.P., G.B., F.A.N.F., and M.O. Laboratory work was performed by R.C., S.Z., and S.F. Analysis was designed and conducted by R.C and B.M. Manuscript elaboration was done by R.C., L.R.D., M.G.C., M.H., Y.L.M., M.O., P.R., N.C.S., E.T., C.L.B., and B.M. Manuscript revision was done by all authors. All authors read and approved the final manuscript.

## Competing interests

The authors declare competing interests.
