## [Transparent Peer Review file · Nature Communications]

Lifestyle change accelerates epigenetic ageing in King penguins

Corresponding Author: Dr Robin Cristofari

Version 1:

Reviewer comments:

Reviewer #2

(Remarks to the Author)

I thank the authors for the clarifications provided. While my main expertise does not lie in mathematical modeling, the explanations given in response to the issues raised by other reviewers appear sound. Similarly, while it is unfortunate that other (metabolic) health metrics – some of which could also be assessed from blood samples – as well as transcriptional data are entirely missing, I understand that obtaining such data can be both technically and logistically challenging. Some remaining comments from my side:

1. I want to stress again a point that might not have come across as clearly as I wanted previously. Looking at the broad picture, CR is one of the primary reasons for choosing this species as a novel model organism, as it is nicely described in the text. However, because the transition to zoos necessarily comes with so many other lifestyle changes, this leaves us in a situation where nothing can actually be concluded with respect to CR per se. Now, this does by no means delegitimize the work shown – I do think, there is value in introducing this model. As far as I can tell, there are no false claims made regarding this, either. But reading through the introduction, which specifically points out the debate around CR effects in long-lived species, sets some expectations, which – to me – are not met. This is an issue of framing and could be resolved by (for example) addressing this limitation directly in the Discussion. Other than that, I support the effort and the prospect of testing this by reintroducing fasting periods at the zoo.

2. The distribution of sampling dates in the wild (now Supplementary Figure 1.) does indeed show a broad distribution, but I would still argue that the main part of the samples was collected somewhat closely to the end of the fasting period. While the epigenome is of course very dynamic, DNA methylation is still relatively stable. In that sense, it would be interesting to see how different the methylation patterns would look, were the samples collected at a time point shortly before the fasting period similar to Zoo Zurich. I assume the younger birds that do not engage in breeding in the wild will also not undergo the associated fasting.

3. I assume that for the zoo-housed animals, there must be some health data available based on veterinary care. Is there no way (with existing data) to validate the relevance of EAA for physical wellbeing in this new model. Could that be shown or commented on.

Reviewer #3

(Remarks to the Author)

The authors have responded in detail to the comments of the reviewers. I remain interested in the dataset and the findings. On the one hand, despite the extensive responses, I strongly believe that the author's preferred approach to a clock is incorrect. It is highly prone to overfitting and it is not truly replicated here (no animals in the wild). When using a biased approach to model building (assume differences exist), much of the statistical arguments used in response to reviewer one are poor substitutes to the standard in the field - replication in a different dataset. As an aside, the discussion about linear vs. logarithmic makes a wrong assumption – that all CpG sites behave the same with aging. They just don't – the "logarithmic"

model applies best to loci that change during development. Many loci (e.g. promoter CpG islands) don't change much with development and instead show very clearly linear methylation change with age.

On the other hand, I applaud the author's unbiased approach based on average CpG methylation (page 8 of the response). In my opinion, the fact that age acceleration in this analysis is much lower than previously found (2.6 years vs. 6.5 years) is not due to lower sensitivity but to lower proneness to overfitting. I would still like to hear more about this approach and its results – for example, why the focus on “negative correlation with age”? Some datasets suggest that loci that gain methylation with age have a different biology (and perhaps higher accuracy for aging) than loci that lose methylation with age (more prone to cell type effects). I also don't fully understand the rescaling approach – if loci are selected based on R values, why does the average value matter and why does it need to be rescaled? One can directly fit a linear model to estimate a methylation age and then calculate age error in each individual – predicted to be positive in the wild and negative in the zoos. As a bonus, this approach allows the authors to much better define the most relevant set of aging loci to dissect the differences between zoo and wild animals (CpG islands? Promoters? Repetitive elements?). I would strongly suggest that Fig. 1 of the paper should be based on this unbiased approach rather than the biased (and possibly overestimating) method first used.

The aDMS PCA (page 17 of the response) is interesting but should be labelled by age (or decade). Have the 4 outliers in this PCA (top right) been excluded from subsequent analyses? I would also suggest showing a PCA of all loci with decent coverage before showing the aDMS PCA. Does an “unselected” PCA show effects of aging/zoo etc.?

Responses to other issues raised are appropriate.

Reviewer #4

(Remarks to the Author)

The authors have clearly and satisfactorily addressed all of the comments I raised in the previous round of review.

Reviewer #5

(Remarks to the Author)

To the Authors,

Thank you for your diligent work and for the detailed responses to the reviewer's comments. I have carefully considered your explanations, particularly regarding the interpretation of epigenetic age acceleration (EAA) and your methodological validations. While I appreciate the effort, I remain unconvinced by certain aspects, and I believe further clarification and additional data are essential to fully support your conclusions.

My primary reservations revolve around three key areas:

1. Interpretation of Intercept-Only Differences in Epigenetic Age: You propose that the observed intercept difference, rather than a slope difference, aligns with a logarithmic model where a constant acceleration factor manifests as an intercept shift. However, my concern persists regarding its biological interpretation within the same species under different conditions. If a true and sustained difference in the rate of biological aging exists due to lifestyle, it should fundamentally lead to a divergence in aging trajectories, reflected as a slope difference over time. The parallel lines, even if mathematically consistent with a logarithmic model, suggest a baseline offset established early, raising strong suspicions of subtle, uncaptured batch effects related to sample origin or handling, despite your rigorous efforts in sample distribution and covariate adjustment.
2. Lack of Early-Life Data and Implications for Non-Linear Analysis: Your acknowledgment that the logarithmic approximation "is not correct for very early life" and that "these early ages are not sampled at all at the zoo" is a critical point. If the intercept difference is indeed a result of rapid initial acceleration in early development, then the most crucial data for observing and accurately modeling this phenomenon is missing. Without epigenetic data from very young animals in both environments, any non-linear analysis of aging trajectories remains speculative. This limits definitive conclusions about the precise timing and impact of early-life lifestyle factors on epigenetic aging.
3. Clarity of Training and Test Sets: I find the explanation regarding the training and testing of your epigenetic clock model to be insufficient. While leave-one-out cross-validation (LOOCV) is a valuable internal validation technique for assessing model performance within a given dataset, it does not substitute for a truly independent, external test set. Without demonstrating the model's predictive power on a completely unseen and separate dataset (e.g., new samples collected independently), the analysis, particularly the significance of the age acceleration plots, remains less convincing. Furthermore, a shared training set might obscure a true difference across these groups; a more revealing approach might involve training a clock exclusively on either the zoo or wild samples and then applying it to the other group. It is imperative to clearly delineate the training and test sets used, and if a fully independent external test set was not employed for the penguin data, this should be explicitly stated as a limitation.

In conclusion, while I commend the innovative approach of using King penguins as a model, these points need critical attention. I strongly recommend providing explicit clarification and transparency regarding the training and test sets used for the penguin epigenetic clock, and generating and incorporating epigenetic data from very young animals (birth to early developmental stages) from both wild and zoo populations. This is crucial for a robust non-linear analysis and to definitively support your interpretation of age acceleration. Until these fundamental data gaps and methodological clarifications are addressed, definitive comments on the causal relationship between lifestyle and epigenetic age acceleration remain premature and potentially an overstatement.

Sincerely,

Version 2:

Reviewer comments:

Reviewer #2

(Remarks to the Author)

We thank the authors for their clarifications. With respect to the distribution of sampling dates, if I understand correctly, the bimodal distribution of sampling of the zoo group could potentially still obscure subtle effects. That being said, this effect (should it exist) would likely be minimal. The proposed addition to the Discussion seems therefore appropriate. With that, all my comments have been addressed.

Reviewer #3

(Remarks to the Author)

I have no further suggestions.

Reviewer #7

(Remarks to the Author)

Cristofari compared epigenetic aging trajectories between king penguins housed in zoos to those from wild animals. This is a unique project with a rarely studied animal species that has the potential to reveal new insights into how lifestyle factors accelerate, or slow down aging processes and the DNA methylation based epigenetic clocks of aging. One important aspect of this work that I didn't see much emphasized is that the authors employed whole genome methylation sequencing (EM-seq) rather than the much more commonly used Illumina arrays which have limited genome coverage. I found the data in this manuscript interesting and convincing. The authors provided satisfactory responses to the reviewers' comments including those made by reviewer #5.

Reviewer #2 (Remarks to the Author):

I thank the authors for the clarifications provided. While my main expertise does not lie in mathematical modeling, the explanations given in response to the issues raised by other reviewers appear sound. Similarly, while it is unfortunate that other (metabolic) health metrics – some of which could also be assessed from blood samples – as well as transcriptional data are entirely missing, I understand that obtaining such data can be both technically and logistically challenging. Some remaining comments from my side:

1. I want to stress again a point that might not have come across as clearly as I wanted previously. Looking at the broad picture, CR is one of the primary reasons for choosing this species as a novel model organism, as it is nicely described in the text. However, because the transition to zoos necessarily comes with so many other lifestyle changes, this leaves us in a situation where nothing can actually be concluded with respect to CR per se. Now, this does by no means delegitimize the work shown – I do think, there is value in introducing this model. As far as I can tell, there are no false claims made regarding this, either. But reading through the introduction, which specifically points out the debate around CR effects in long-lived species, sets some expectations, which – to me – are not met. This is an issue of framing and could be resolved by (for example) addressing this limitation directly in the Discussion. Other than that, I support the effort and the prospect of testing this by reintroducing fasting periods at the zoo.

We agree with this important nuance, and have added the following to our discussion (lines 302-307): *“Naturally, the changes associated with the transition to zoo conditions are broad and multifaceted, extending beyond reduced caloric intake and physical activity. While this study demonstrates the impact of this transition on ageing rate, and suggests that metabolic regulation is deeply involved, the precise mechanisms through which this impact happens remain to be explored - whether through CR, sedentary behaviour, disruption of life rhythms, reduced environment complexity, altered microbial environment, or other factors.”*

2. The distribution of sampling dates in the wild (now Supplementary Figure 1.) does indeed show a broad distribution, but I would still argue that the main part of the samples was collected somewhat closely to the end of the fasting period. While the epigenome is of course very dynamic, DNA methylation is still relatively stable. In that sense, it would be interesting to see how different the methylation patterns would look, were the samples collected at a time point shortly before the fasting period similar to Zoo Zurich. I assume the younger birds that do not engage in breeding in the wild will also not undergo the associated fasting.

Thank you for raising this interesting point. We are currently engaged in answering this question through experimental fasting in zoo settings. Unfortunately, our wild sampling efforts are currently suspended due to the highly pathogenic avian influenza epizootic, but we hope to resume them as soon conditions permit.

In the meantime, the circannual cycle is indeed known to be reflected in birds' methylome, as shown for example by Lindner et al. (2020). While our dataset was not explicitly designed to capture this, we do observe seasonal variation. Specifically, we use elastic net regression to relate the day of the season (defined as October 1st for the Southern Hemisphere and April 1st for the Northern Hemisphere) to the principal components derived from the whole-methylome PCA. The day of the season can be accurately predicted based on the methylation variation (with a pseudo-R-squared of 0.89, see panel A). However, the total amount of CpG methylation variance explained is only 4.97% (calculated as the product of the squared elastic net regression weights by PCA eigenvalues, summed over all PCs, and divided by the sum of eigenvalues). Moreover wild and zoo birds do not significantly differ along this seasonal axis (panel B), and “origin” does not significantly contribute to the fitted linear model.

In summary, we find that (1) indeed, the point in the breeding cycle at which a bird is sampled leaves a signature in that bird's methylome (as expected), (2) that seasonal signal reflects only a minor part of the total variance in CpG methylation, and (3) there is no evidence for a systematic bias in our samples due to sampling dates between wild and zoo groups. However, we added the following to our discussion (lines 316-320): *“the sampling dates are not entirely standardised in our study relative to the onset of the breeding season. All samples have been collected during the breeding season, and dates do not significantly differ between groups: but the epigenome is known to change with the circannual cycle (Lindner et al. 2020), and this incomplete standardisation may introduce some noise - although likely not a systematic bias - into our results.”*

3. I assume that for the zoo-housed animals, there must be some health data available based on veterinary care. Is there no way (with existing data) to validate the relevance of EAA for physical wellbeing in this new model. Could that be shown or commented on.

This would indeed have been ideal. Unfortunately, such data currently do not exist. Although there is an ongoing effort to log veterinary interventions more systematically in electronic databases, this is far from covering the past 40 (or even 15) years, and veterinary logs do not follow individuals if or when they are transferred between zoos. This lack of systematic recording leads to biases in the data which tend to focus more on certain types or levels of pathology. Consequently, our understanding of the health history of the zoo-housed penguins in this study is limited to anecdotal information. This project has actually facilitated discussions about this shortcoming, and we hope that future initiatives will enable more systematic and comprehensive health data collection.

Reviewer #3 (Remarks to the Author):

The authors have responded in detail to the comments of the reviewers. I remain interested in the dataset and the findings.

On the one hand, despite the extensive responses, I strongly believe that the author's preferred approach to a clock is incorrect. It is highly prone to overfitting and it is not truly replicated here (no animals in the wild). When using a biased approach to model building (assume differences exist), much of the statistical arguments used in response to reviewer one are poor substitutes to the standard in the field - replication in a different dataset.

We agree that replication in a different dataset would be the “gold standard” to demonstrate the robustness of our results. However, as in many studies, particularly those relying on genomic data, there are limitations on the available sample size: allocating samples to a test set reduces the power on training. This constraint guided our choice of using a leave-one-out approach for training/test design (i.e., 63 training samples and one test sample for each point). We acknowledge that this approach cannot fully

substitute for validation in a completely independent wild population. As we have emphasized in our previous response, however, a different known-age dataset in the wild simply does not exist. We made every effort to minimize batch effects, as discussed in earlier responses, but ultimately, we agree that, as in many cases, only a fully independent replication of this study will settle this point entirely.

Thanks to a suggestion made by Reviewer 5, we also provide an additional form of independent validation of our approach. Specifically, we trained the epigenetic clock model solely on all zoo samples, and then used it to predict the ages in all wild samples, or vice versa. These two approaches yield very similar results, and are highly consistent with our main analyses (see below for details, and Supplementary Figure 7 along with the accompanying text). While this is not a substitute for a full replication study, we believe it will address the concerns about potential overfitting. This method, similar to the mean-aDMS approach, should be entirely immune to group overfitting.

We now explicitly acknowledge this limitation in the discussion (lines 311-323): *“This study has three additional methodological limitations: [...] Third, we include two zoos, but only one wild population - as there is currently only one known-age wild King penguin population covering the whole age span of King penguins in the world (Bardon et al. 2023). This study thus lacks a fully independent biological replicate in the wild.”*

As an aside, the discussion about linear vs. logarithmic makes a wrong assumption – that all CpG sites behave the same with aging. They just don’t – the “logarithmic” model applies best to loci that change during development. Many loci (e.g. promoter CpG islands) don’t change much with development and instead show very clearly linear methylation change with age.

This may be a misunderstanding: we do not make the assumption that all CpG sites behave the same with aging. We fully agree with the reviewer that some loci have a marker logarithmic trajectory, some behave locally linearly in adulthood, and some follow apparently stochastic trajectories. In theory, however, it is impossible for methylation at any single CpG to be truly linear with age: methylation levels are bounded between 0 and 1, but a strictly linear change would imply the existence of methylation values outside of [0,1]. We point to a recent article by Ochana and colleagues (Cell Reports 2025, 10.1016/j.celrep.2025.115958), which provides convincing examples at the CpG level.

Here we rather make the point that *epigenetic age*, which is a methylome-wide index, follows a logarithmic relationship with time. This general behaviour of epigenetic clocks has been well documented, as we briefly mentioned in our previous answer. We agree that this logarithmic pattern is most prominently observed during development (it was initially thought to be primarily relevant before age 20 in humans, see Horvath et al. 2013). This is largely because the logarithmic curve becomes harder to distinguish from a straight line, when we focus on a small range of larger predictor values (typically, a range of adult ages).

This actually justifies our choice of model, when we approximate epigenetic age by a straight line in our analysis, in full agreement with the reviewer’s perspective. Our point about the logarithmic model is less relevant for interpreting the shape of the EA curve, that interpreting the intercept shift, that we discussed in our previous revision.

On the other hand, I applaud the author’s unbiased approach based on average CpG methylation (page 8 of the response). In my opinion, the fact that age acceleration in this analysis is much lower than previously found (2.6 years vs. 6.5 years) is not due to lower sensitivity but to lower proneness to overfitting.

We thank the reviewer once again for suggesting that we include this simpler approach in the first place! Indeed, we agree it provides stronger support for the main result. Regarding the magnitude of the observed effect, we do agree this remains an open point for discussion. On one hand, the smaller, more

conservative estimate is a reliable “lower bound” for the effect of zoo lifestyle, which we now underline in our manuscript.

On the other hand, in our human test data, our approach produces estimates that closely align with the well-established PCPhenoAge and PCGrimAge clocks (see Supplementary Figure 8 C-D and accompanying text: “*Final mean difference between groups was 11.8 years according to our model (SE = 1.24, p-value < 1e-12), to be compared with 14.0 years according to PCPhenoAge (SE = 0.64, p-value < 1e-16) and 11.6 years according to PCGrimAge (SE = 0.57, p-value < 1e-16).*”) This does not suggest our model does not overfit / overestimate inter-group differences. Furthermore, the mean-aDMS approach aggregates sites with equal weights, including strong and weak predictors of age.: We expect it to have a somewhat lower predictive power compared to models that incorporate a larger part of covariance with age. For these two reasons, we think the 6.5 years estimate is likely to be accurate in this case.

In both cases, the estimates lead to a qualitatively identical conclusion, and we suggest to provide both, together with interpretative context, in our manuscript at lines 143-149:

“The numerical differences between these approaches may be explained by different sensitivities (training sample size is halved in the first of these controls, and the second one is based a cruder averaging procedure that gives equal weights to all included aDMS CpGs, regardless of the strength of their actual covariance with age).. The true value of the age acceleration at the zoo is thus likely between 2.63 years (the most conservative estimate) and 6.48 (the estimate derived from a PCA of the whole sample set) - in any case, we reach the conclusion that zoo-housed birds are significantly age-accelerated compared to their wild counterparts.”

We also amended the numbers in our discussion accordingly at lines 330-331.

I would still like to hear more about this approach and its results – for example, why the focus on “negative correlation with age”? Some datasets suggest that loci that gain methylation with age have a different biology (and perhaps higher accuracy for aging) than loci that lose methylation with age (more prone to cell type effects).

We appreciate the reviewer's insightful comment regarding the focus on loci with a negative correlation with age. We fully agree with the reviewer that CpGs with positive and negative correlations with age should not be considered as biologically homogeneous. Indeed, Horvath (2013, Genome Biology) already proposed that loci with positive correlation to age tend to exhibit less variance across tissues, although this effect was most pronounced in fetal tissues rather than in adults) More recent studies (Moqri et al. 2024, Nat. Com.) have convincingly demonstrated that polycomb repressive complex 2 (PRC2) targets consistently gain methylation with age and serve as reliable epigenetic aging biomarkers .

In our original PC-elastic-net analysis, we included both positively and negatively-correlated CpGs. However for the “validation” analysis we opted to focus solely on negative correlation loci. This choice was driven by the distribution of CpGs in the King penguin genome, where only 366 CpGs exhibit a strong positive Pearson correlation with age, compared to 9,839 with negative correlation. This is notably different compared with Horvath's original clock, where > 50% of clock-sites show positive correlations with age. These CpGs are not all independent either, for example, 94 CpGs with positive correlation with age are grouped in just 4 loci. And while it would have been extremely interesting to replicate e.g. the PRC2-AgeIndex approach described by Moqri et al (2024, Nat. Com.), we lack direct PRC2 binding information in this non-model species. Hence, in our case, conclusions derived from these comparatively few positive-correlation sites would unfortunately not be robust.

This being said, this difference from human data merits further exploration and could provide valuable insights into species-specific epigenetic ageing mechanisms.

I also don't fully understand the rescaling approach – if loci are selected based on R values, why does the average value matter and why does it need to be rescaled? One can directly fit a linear model to estimate

a methylation age and then calculate age error in each individual – predicted to be positive in the wild and negative in the zoos.

We apologize for any lack of clarity in our previous explanation - this is actually what we did in the previous revision. To clarify, the rescaling was not applied at the individual CpG level, but was performed on the average methylation of the selected aDMS, prior to the linear modelling. This rescaling is performed to provide model coefficients that are directly interpretable, with age acceleration expressed in units compared to chronological age. We agree that in this case, this rescaling could also have been performed on the coefficients after the linear regression without loss of information.

We have edited the main text to make this clearer (lines 136-137):

“The average methylation value across aDMS sites was linearly scaled to the observed range of ages in our data, and linearly modelled as a function of log(age)”

As a bonus, this approach allows the authors to much better define the most relevant set of aging loci to dissect the differences between zoo and wild animals (CpG islands? Promoters? Repetitive elements?). I would strongly suggest that Fig. 1 of the paper should be based on this unbiased approach rather than the biased (and possibly overestimating) method first used.

We are not sure we entirely understand the suggestion. On the one hand, using the full methylation matrix as a predictor in a single model would lead to a problem of degrees of freedom. It is precisely to avoid this that we used feature preselection in our original analysis (based on Pearson coefficient and elastic net regression, which is a standard choice in the field). On the other hand, fitting one model per CpG site (an Epigenome-Wide Association Study (EWAS)-like approach) is possible: that is what we did for feature pre-selection. But it is unclear how the individual-level residuals (not per-CpG, per-individual level) are obtained in that case. While we agree developing the EWAS aspect further is an extremely interesting approach, we believe it would not directly add to this study, which focuses on the difference in ageing rate between lifestyle groups, not on the impact of ageing itself on the epigenome.

In this revised version, we have included the approach suggested above by the reviewer (a linear model of average aDMS methylation as a function of age), together with our original approach, in the main figure. We think they provide complementary and consistent lines of support for our conclusions - and we thank the reviewer again for suggesting this additional validation method in the first place, which strengthens the robustness of our findings!

The aDMS PCA (page 17 of the response) is interesting but should be labelled by age (or decade). Have the 4 outliers in this PCA (top right) been excluded from subsequent analyses? I would also suggest showing a PCA of all loci with decent coverage before showing the aDMS PCA.

We have updated this figure to include age labels for each bird, which clarify the relationship between the PCA positions and chronological age. Additionally, we now present both PCA analyses with the entire methylome (filtered for coverage as explained in the main methods, panel A), and aDMS-only (panel B), allowing for clearer comparison. The 4 “outliers” visible on panel B are actually the 4 young birds that are also clearly apparent on Fig. 1A of the manuscript. In regard to aDMS PC1 (panel B), these are actually not outliers *stricto sensu*: their position is as expected given the age distribution, and very consistent with their average methylation at aDMS (see Fig. 1C in the main text). Thus, they have not been excluded from any analysis. We acknowledge a sampling gap in the wild birds between approximately 1 and 2.5 years of age corresponding to the juvenile “wandering years”, when birds are typically absent from the colony. In the zoo, these ages could not be sampled at all due to a lack of recent births at the time of the experiment.

Does an “unselected” PCA show effects of aging/zoo etc.?

It does indeed: this effect was shown on Fig. S6E of the previous version of this manuscript and commented in the accompanying text. Briefly, the first PC of a whole-methylome PCA reflects both age and origin (linear model: ΔAIC with the null model: 33.8, p-value for the “origin” factor = 1.07×10^{-8}), but only explains a limited amount of the total variance (4.90%). As we noted there, however, “we underline that in neither cases a principal component is guaranteed to covary with age (this is not e.g. an RDA), and PC1 is extremely likely to carry other signals than age.”

Responses to other issues raised are appropriate.

Reviewer #4 (Remarks to the Author):

The authors have clearly and satisfactorily addressed all of the comments I raised in the previous round of review.

We thank the reviewer for their kind words!

Reviewer #5 (Remarks to the Author):

To the Authors,

Thank you for your diligent work and for the detailed responses to the reviewer's comments. I have carefully considered your explanations, particularly regarding the interpretation of epigenetic age acceleration (EAA) and your methodological validations. While I appreciate the effort, I remain unconvinced by certain aspects, and I believe further clarification and additional data are essential to fully support your conclusions.

My primary reservations revolve around three key areas:

1. Interpretation of Intercept-Only Differences in Epigenetic Age: You propose that the observed intercept difference, rather than a slope difference, aligns with a logarithmic model where a constant acceleration factor manifests as an intercept shift. However, my concern persists regarding its biological interpretation within the same species under different conditions. If a true and sustained difference in the rate of biological aging exists due to lifestyle, it should fundamentally lead to a divergence in aging trajectories, reflected as a slope difference over time. The parallel lines, even if mathematically consistent with a logarithmic model, suggest a baseline offset established early, raising strong suspicions of subtle, uncaptured batch effects related to sample origin or handling, despite your rigorous efforts in sample distribution and covariate adjustment.

We thank the reviewer for raising this crucial point. Although we agree it runs counter to intuition, sustained differences in the rate of biological ageing should *not* lead to slope differences over time under the logarithmic ageing model (i.e. under the model where epigenetic age changes by a fixed amount for each doubling or other *n*-ing of chronological age), provided both groups are observed in the same chronological age (*x*-values) range. This divergence would be observed if the *rate of acceleration of ageing*, as opposed to the *rate of ageing*, was different between the two groups.

We illustrate this in the figure above: panel A illustrates the simple logarithmic ageing model proposed by Alisch et al. 2012, Horvath et al. 2013, and formalised in Snir et al. 2019. Panel E shows the (constant) intercept difference between these two curves. Despite the strong optical illusion, **the difference between the two log lines at any given age is constant. It does not increase faster in early life, nor is it required to be established in early life.** In order to follow the log-intercept model, we simply hypothesise

that the difference in average ageing rate is approximately constant throughout the follow-up period in each group.

Panels B-D show the more general model where the rate of ageing is itself a function of time (any arbitrary function is possible - a constant as in A, linear with time as in B, exponential in C, logistic in D - or anything else). In that case, the log curves can indeed diverge: in the linear acceleration case shown here (as in any other monotonous acceleration), the difference between the groups indeed increases with time (panels F-H). **It is not impossible that this more general model would actually be selected with a much larger dataset**, especially if it involved e.g. a small linear rate increase (the B case). In our case, it is rejected by AIC - but it would not be contradictory with our findings. Here, we simply suggest that the log-intercept parameter accounts for most of the observed variance, and that in comparison with it, the difference in average slope is too small to be accurately fitted in a dataset of “only” 64 individuals.

In the edge case where that arbitrary rate function is the exponential (panel C), we indeed would revert back to diverging lines, i.e. to the linear model of ageing. The difference between groups would increase linearly (panel G). Thus, the linear model can actually be understood as an extreme edge case of the logarithmic model, under which the fundamental measure of time remains logarithmic (each doubling of life-time marks a unit of ageing) but the rate of ageing increases exponentially.

Naturally, these observations only hold if epigenetic ageing is indeed logarithmic with time - but the fit of our data is rather unambiguous here. **Importantly, we refer to the “smoke test”** included in our study. In this case, smoking is not an early-life stressor, and the subjects included in the study were current smokers or current non-smokers. The stressor (tobacco) is maintained for an extended period whose duration is not included in the model. In that case too, this results in parallel EA lines (Sup. Fig. 7): yet it is **difficult to argue that this is the result of a batch effect or of an unmeasured early life stressor, and not simply of tobacco smoking**. This case is well represented as the logistic acceleration model (panels D and H): ageing rate is initially very similar between groups, but increases sharply in the stressed group when the stress (e.g. tobacco use) begins (the curve divergence period), and settles at a higher value. When the stressor is stable, the logarithmic curves are parallel again (panel H).

In summary, **we do not suggest there is no variation (acceleration) in ageing rates in either of the two groups**. We simply state that we do not observe this variation in our sample (possibly because it is too small to be fitted here), and that this is not contradictory with our conclusions. In fact, **this represents the “null model” where ageing rate difference is close to constant** over the observation period. More complex models could definitely be explored in the future with the appropriate (longitudinal) experimental design!

2. Lack of Early-Life Data and Implications for Non-Linear Analysis: Your acknowledgment that the logarithmic approximation "is not correct for very early life" and that "these early ages are not sampled at all at the zoo" is a critical point. If the intercept difference is indeed a result of rapid initial acceleration in early development, then the most crucial data for observing and accurately modeling this phenomenon is missing. Without epigenetic data from very young animals in both environments, any non-linear analysis of aging trajectories remains speculative. This limits definitive conclusions about the precise timing and impact of early-life lifestyle factors on epigenetic aging.

We entirely agree. However, here, we do not draw any conclusions about the precise timing and impact of early-life factors, nor do we suggest that the difference in ageing rate observed in adulthood is the *result* of early life factors. If we were to speculate, we would lean for model D in the figure above, with a gradual establishment of ageing rate difference in the first years of life, as lifestyle differences will be more marked from fledging onward, when wild birds would start to actively swim in the open ocean before starting the breeding life (and associated fasting), while zoo birds remain sedentary and well fed. But **this would not mean that the difference is a result of early life events**. It would mean that **the difference starts to occur during or after early life**. This is a key difference, and best exemplified here again by the smoker test: the

parallel curves observed in adulthood in Supplementary Figure 8A are not a result of whatever happened *before* the smoking, but of the ongoing difference in smoking itself.

We would also like to nuance the idea that the logarithmic approximation is not correct for very early life. Our point was that the logarithmic approximation *is* correct, but that our representation of the logarithmic curve by a linear one is probably not accurate for the very early life stages.

We also agree that studying early life effects would be fascinating. This would involve dense longitudinal sampling in a sufficient number of chicks from hatching onward, both at zoos and in the wild. While this is clearly outside the scope of this study, this would indeed yield fascinating insights into early life determinants of age acceleration!

We have clarified these points in the manuscript at lines 311-316: *“This study has three additional methodological limitations: first, we focus mainly on adult, breeding-age birds. Thus, we are not able to examine whether the apparent difference between wild and zoo-housed birds is established already in early life, or appears gradually. While the intercept difference between the two groups (Fig. 1A) is very consistent with logarithmic EA in presence of a stable age-accelerating stressor (as developed in Supplementary Methods S4), we cannot exclude strong effects of early-life rearing conditions.”*

3. Clarity of Training and Test Sets: I find the explanation regarding the training and testing of your epigenetic clock model to be insufficient. While leave-one-out cross-validation (LOOCV) is a valuable internal validation technique for assessing model performance within a given dataset, it does not substitute for a truly independent, external test set. Without demonstrating the model's predictive power on a completely unseen and separate dataset (e.g., new samples collected independently), the analysis, particularly the significance of the age acceleration plots, remains less convincing. Furthermore, a shared training set might obscure a true difference across these groups; a more revealing approach might involve training a clock exclusively on either the zoo or wild samples and then applying it to the other group. It is imperative to clearly delineate the training and test sets used, and if a fully independent external test set was not employed for the penguin data, this should be explicitly stated as a limitation.

We agree with the reviewer that adding additional groups to this study would have been ideal. Unfortunately, there are also some necessary practical trade-offs that prevent the acquisition of an unlimited number of samples. However, training the clock on one group only and predicting for the other group, as suggested by the reviewer, is one way to alleviate this: it also robustly supports our conclusions. We show the result of this test as a new Supplementary figure 7, reproduced here below.

Briefly, we used glmnet to fit an elastic net model to our aDMS PCA coordinates, using LOOCV (as in our main analysis), but using only half of the birds to train the model: either zoo birds only (A) or wild birds only (B). Each time, the other group was predicted from that model. We show the predicted values (“Epigenetic age”) regressed against the chronological age, together with the coefficients of the corresponding linear model ($EA \sim \text{age} + \text{origin}$). The null model is simply $EA \sim \text{age}$. In this case, the intercept difference for origin is fully absent from the training data, and is only inferred when regressing the predicted values against age.

In both cases, estimates of age acceleration are well supported by AIC, strongly significant, and larger than in the mean-aDMS approach. Training on zoo birds only yields a value close to our original estimate (6.00 vs 6.48 years), and training on wild birds only yields an intermediate value (3.83 years). However, while this approach provides more independent support for our conclusions, we underline the fact that it relies on a halved sample size for model training, and is likely to produce noisier coefficient values - this motivated our initial choice of methods.

We have now mentioned this additional validation in the main text at lines 128-133, and to our Supplementary methods (lines “Additional validations - c” and Supplementary Figure 7).

We also now explicitly stated this limitation point of debate:

“The numerical differences between these approaches may be explained by different sensitivities (training sample size is halved in the first of these controls, and the second one is based a cruder averaging procedure that gives equal weights to all included aDMS CpGs, regardless of the strength of their actual covariance with age). The true value of the age acceleration at the zoo is thus likely between 2.63 years (the most conservative estimate) and 6.48 (the estimate derived from a PCA of the whole sample set) - in any case, we reach the conclusion that zoo-housed birds are significantly age-accelerated compared to their wild counterparts.”

Regarding the delineation of training and test sets in our original analysis, our manuscripts states that “For each grid value, we performed a leave-one-out fit: the age of zoo individuals was first transformed linearly using the local slope and intercept, and, taking each sample out in turn, we trained an elastic net model using all principal components of methylation variance using N-fold internal cross-validation on the 63 remaining samples. We then used this model to predict epigenetic age for the 64th sample”. This is a full description of our training / test design: for each point, the other 63 points are the training set. We underline that this is independent from the LOOCV procedure used to train the model on these 63 points. We have now added an additional clarification to our methods at lines 405-406, immediately after the note cited here: “In other words, for each point, the other 63 points are the training set, and LOOCV is performed among these 63 training points, before predicting EA for the 64th point.”

In conclusion, while I commend the innovative approach of using King penguins as a model, these points need critical attention. I strongly recommend providing explicit clarification and transparency regarding the training and test sets used for the penguin epigenetic clock, and generating and incorporating epigenetic data from very young animals (birth to early developmental stages) from both wild and zoo populations. This is crucial for a robust non-linear analysis and to definitively support your interpretation of age acceleration. Until these fundamental data gaps and methodological clarifications are addressed, definitive comments on the causal relationship between lifestyle and epigenetic age acceleration remain premature and potentially an overstatement.

We hope that this revised version alleviates these concerns. We agree with the reviewer that studying early life stages at high resolution will be a fascinating and highly informative endeavour, and will likely

constitute a complete follow-up project. At this stage, this remains challenging at the zoo (fecundity rates are low and it would take years to gather a sufficient sample size, which precludes gathering such data within the framework of this review process). However, we hope we made it clear now that we do not draw any conclusions regarding non-linearity in early age in this study. We thank the reviewer for their constructive suggestions that have allowed us to provide additional and independent support for our results!